# Co-GAIL: Learning Diverse Strategies for Human-Robot Collaboration

**Chen Wang, Claudia Pérez-D'Arpino, Danfei Xu, Li Fei-Fei, C. Karen Liu, Silvio Savarese**
Stanford University

**Abstract:** We present a method for learning human-robot collaboration policy from human-human collaboration demonstrations. An effective robot assistant must learn to handle diverse human behaviors shown in the demonstrations and be robust when the humans adjust their strategies during online task execution. Our method co-optimizes a human policy and a robot policy in an interactive learning process: the human policy learns to generate diverse and plausible collaborative behaviors from demonstrations while the robot policy learns to assist by estimating the unobserved latent strategy of its human collaborator. Across a 2D strategy game, a human-robot handover task, and a multi-step collaborative manipulation task, our method outperforms the alternatives in both simulated evaluations and when executing the tasks with a real human operator in-the-loop. Supplementary materials and videos at https://sites.google.com/view/cogail/home

**Keywords:** Learning for Human-Robot Collaboration, Imitation Learning

## 1 Introduction

Advancing technologies for human-robot collaboration (HRC) has the potential to unlock applications with large societal impact in manufacturing, hospitals, and home settings [1, 2]. However, robots that are designed to work around humans are still limited in versatility when performing collaborative tasks. Recent advances in robot learning focus on robots that work in isolation [3, 4, 5, 6] or alongside other agents that do not exhibit human traits [7, 8, 9, 10]. Learning to collaborate with humans presents unique challenges to existing robot learning methods: instead of optimizing only for efficient task completion, the robot assistant must act in coordination and adapt to the diversity of strategies and movements of their human counterparts. This work aims to develop robot assistants that adapt to diverse human strategies and movements in collaborative manipulation tasks.

While past works in multi-agent reinforcement learning show that agents could learn to develop collaborative behaviors [7, 9, 11, 8], hand-engineering reward functions for collaborative goals is non-trivial [10] and the strategies learned from robot-robot collaboration may not align with that of the human users. On the other hand, learning from demonstrations of human-human collaboration could allow the robot to develop plausible collaboration strategies, as well as avoiding reward engineering. However, most prior works in this setting [12, 13, 14] treat the human as a stationary part of an environment during the learning process and fail to model the human's reactions to the evolving robot policy. Modeling such reciprocal behavior adjustment during training is essential to develop robust assistive policies, as the human collaborator are likely to depart from the exact strategies they previously demonstrated during online task execution. In addition, human behaviors tend to be highly diverse both in high-level intents and low-level movements. Learning a robot policy that operates effectively with diverse human behaviors is also crucial for HRC tasks.

In this work, we propose a method to learn policies that generate diverse human and robot collaboration behaviors from human-human demonstrations with an interactive *co-optimization* process. Instead of treating humans as part of the environment, the human policy is an active participant in the training process and co-evolves with the robot policy as the learning progresses. This allows the robot policy to adapt to the movements of the human teammate while assisting the human to complete the task. To overcome the challenge of diverse strategies presented in the human demonstrations, we introduce a latent representation of human collaboration strategies. On one hand, our latent strategy learning objective allows the policy to discover and disentangle factors of variations in the demonstrations, ensuring that the robot assistant is proficient in all collaborative behaviors

5th Conference on Robot Learning (CoRL 2021), London, UK.

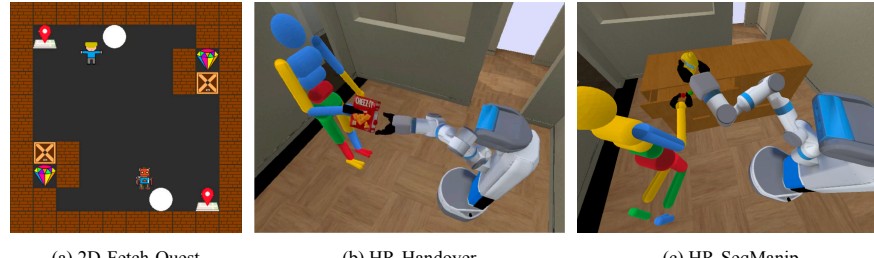

|  (a) 2D-Fetch-Quest | (b) HR-Handover | (c) HR-SeqManip |

Figure 1: **Human-robot collaboration tasks**: (a) 2D-Fetch-Quest: a human plays with a learned agent to unlock and retrieve objects with strategic dependencies; (b) HR-Handover: human hands over an object to a robot; (c) HR-SeqManip: Pick an object and open a drawer to drop the object on it. Human and robot might lead different steps of the task.

shown by the expert even if certain strategies are rare. On the other hand, the latent strategy code serves as a communication protocol that enables the robots to anticipate future behaviors of their human collaborators and to become more effective during human-in-the-loop task execution.

We test the method in three challenging collaborative domains illustrated in Figure 1: 2D game that requires coordination among two agents (**2D-Fetch-Quest**), human-to-robot handover (**HR-Handover**), and sequential collaborative manipulation with handovers (**HR-SeqManip**). These domains range in difficulty from a 2D grid-like game to high-dimensional manipulation tasks with 7-DOF arms and task execution within a physics-based simulation environment using Bullet [15] and iGibson [16]. We conclude by presenting a brief demonstration of the method in interaction with real human collaborators, showing for the first time a learning method that succeeds at adapting to real users in these challenging tasks.

## 2 Related work

**Multi-agent Reinforcement Learning:** A large body of prior works utilize multi-agent RL (MARL) to learn collaborative strategies for tackling multiplayer games [17, 11], multi-robot manipulation [18, 19], traffic control [20], and social dilemmas [7]. However, in the HRI setting, the robot needs to learn to adapt to a wide range of human collaborative behaviors, for which it is nontrivial to design effective reward functions. Translating the benefits of MARL to HRI in real-world manipulation tasks is limited because agents are assumed to have similar policies or don't exhibit human traits [21]. Tampuu et al. [10] use multi-agent reinforcement learning to model both agents and demonstrate how collaborative behavior can arise with the appropriate choice of reward structure. In [8] the robot learns to influence the collaborator's action by optimizing a long-term reward over repeated interactions and estimating their policy per interaction episode through a learned latent space. Hand-engineered collaborative rewards can cause the overfitting of the learned robot policy [9], limiting the robot's adaption to diverse human behaviors.

**Multi-agent Inverse Reinforcement Learning:** Inverse reinforcement learning (IRL), on the other hand, shows promising results in learning collaborative behaviors learning from expert demonstrations [22, 23]. Notably, recent progress in generative adversarial imitation learning (GAIL) [24, 25] further shows effectiveness in cooperative settings [26]. However, as noted in prior works [25], the learned collaborative policy tends to overfit strategies that are easy to execute and neglects the more challenging cases. This becomes problematic when the robot encounters a real human collaborator, who may prefer strategies outside of the distribution that the policy focuses on. Our method addresses this problem by explicitly disentangling the behaviors shown in the demonstrations and encourage the robot policy to learn from diverse collaborative strategies.

**Modeling diverse behavior:** Learning diverse behaviors has been a long-standing challenge in reinforcement learning research [27, 28, 29]. Recent works show that by maximizing the entropy of states and minimizing the conditional entropy of states with respect to the actions, policies can learn diverse skills [30, 31] even without reward signals [32]. Rather than training policies to behave as diverse as possible, our method optimizes for covering the behaviors shown in demonstrations. This allows the robot to focus on collaborative behaviors that are plausible during human-robot explo-

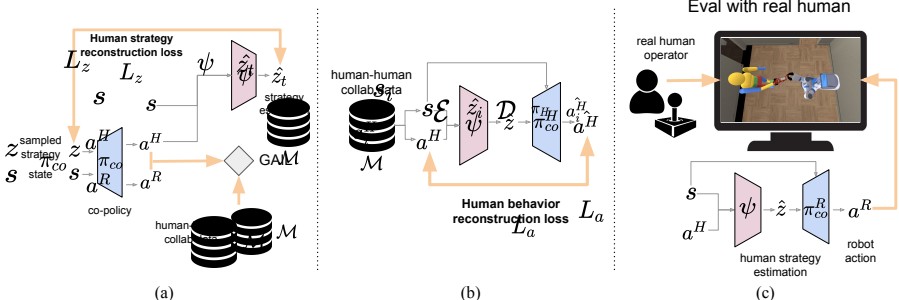

Figure 2: **Architecture Overview**: (a) The co-policy $\pi_{co}$ observes the states $s$ and strategy code $z$ and outputs actions for both human and robot $(a^H, a^R)$. $\pi_{co}$ is trained with the GAIL algorithm using a dataset of human-human collaboration $\mathcal{M}$. The strategy recognition network $\psi$ is jointly trained to estimate the latent strategy from human behavior, using the reconstruction loss $L_z$ to recover $z$. (b) The $\psi$ and $\pi_{co}$ networks are also trained as an an auto-encoder using the behavior reconstruction loss $L_a$ to learn the inverse mapping from the demonstration to the strategy space and encourage the coverage of diverse strategies. (c) System for testing human-robot collaboration. The model estimates the human strategy $\hat{z}$ from its motions and outputs the robot's action $a^R$.

ration, which is especially crucial in large environments where behavioral diversity is unbounded. Clegg *et al.* [33] introduces a method to learn diverse and plausible assistive dressing behavior by hand-coding human motor capabilities in the model. In contrast, our method does not assume explicit domain knowledge and learns collaborative policies from demonstration data. Wang *et al.* [34] proposes to learn diverse behaviors through imitation but requires access to demonstrations during online execution, whereas our method only assumes access to demonstration at training time.

**Human behavior modeling in human-robot collaboration:** Prior works approach the problem of modeling human behaviors in interactive learning settings in multiple ways. Most works introduce intermediate representation of the human behaviors such as user type classification [12, 35], future action prediction [36], hidden goal estimation [37], information communication [38] and human motion prediction [39]. These methods show promising results in HRI applications, including assistive robot for dressing [33, 40], feeding [41, 42, 43], assembling [14] etc. However, most existing works requires tasks-specific knowledge such as user types [12, 35]. Our method learns a latent space of collaboration strategy from demonstrations without explicit task-level supervision.

## 3 Method

Our method is designed to overcome two unique challenges imposed by human-robot interaction: (1) human behaviors are difficult to model because they follow their own policy while reacting to the robot's actions, and (2) human behaviors are highly diverse both in terms of the high-level strategies and the low-level actions. Below we introduce the general HRC problem setup (3.1), an interactive imitation learning formulation to the problem (3.2), and how to uncover diverse collaboration behaviors by learning a latent strategy space (3.3). Finally, we introduce the learning algorithm *Co-GAIL* (Algorithm 1) and the human-in-the-loop task execution setup (3.4).

### 3.1 Learning human-robot co-policy

Although human and robot have different state space and action space, their underlying motivation of completing the collaboration task is shared. Instead of treating the human as a part of the environment with a fixed policy, allowing the human and the robot to learn and explore concurrently has the potential to achieve a more robust robot policy. We consider a human-robot collaboration task as an extension of Markov Decision Process (MDP) called Markov games [44] for two agents $(N = 2)$, $MG = (\mathcal{S}, \mathcal{A}^H, \mathcal{A}^R, \mathcal{T}, R, \gamma, \rho_0)$, with state space $\mathcal{S}$, human action space $\mathcal{A}^H$, robot action space $\mathcal{A}^R$, transition distribution $\mathcal{T}(\boldsymbol{s}_{t+1}|\boldsymbol{s}_t, \boldsymbol{a}_t^H, \boldsymbol{a}_t^R)$, reward function $R(\boldsymbol{s}_t, \boldsymbol{a}_t^H, \boldsymbol{a}_t^R, \boldsymbol{s}_{t+1})$, discount factor $\gamma \in [0, 1)$, and initial state distribution $\rho_0$.

The solution to this MDP is a co-policy $\pi_{co}$ with a shared policy network that outputs action for both agents. At every step, the co-policy observes the current state $\boldsymbol{s}_t$, chooses a set of actions

$(\boldsymbol{a}_t^H, \boldsymbol{a}_t^R) \sim \pi_{co}(\boldsymbol{a}_t^H, \boldsymbol{a}_t^R | \boldsymbol{s}_t)$, and observes the next state $\boldsymbol{s}_{t+1} \sim \mathcal{T}(\cdot | \boldsymbol{s}_t, \boldsymbol{a}_t^H, \boldsymbol{a}_t^R)$ and reward $r_t = R(\boldsymbol{s}_t, \boldsymbol{a}_t^H, \boldsymbol{a}_t^R, \boldsymbol{s}_{t+1})$ shared by both agents. The goal is to learn a co-policy that maximizes the shared expected return $\mathbb{E}[\sum_{t=0}^{\infty} \gamma^t R(\boldsymbol{s}_t, \boldsymbol{a}_t^H, \boldsymbol{a}_t^R, \boldsymbol{s}_{t+1})]$. With a slight abuse of notation, we denote the part of the policy that outputs human actions as $\pi_{co}^H$ and that for robot actions as $\pi_{co}^R$.

## 3.2 Learning strategy-conditioned co-policy from demonstration

While we could attempt to solve for the co-policy using existing RL methods, designing an effective and comprehensive reward function that properly evaluates the quality of human-robot collaboration is non-trivial. On the other hand, imitation learning has the advantage of learning the policy directly from demonstrations, bypassing the need to explicitly design a reward function. As such, we record a set of *human-human* demonstrations $\mathcal{M}$ of two users performing a cooperative task and formulate the co-policy learning problem based on MA-GAIL [26], a multi-agent extension of the Generative Adversarial Imitation Learning (GAIL) [24] framework.

Let $\pi_{E_1}$ and $\pi_{E_2}$ denote two expert policies to which we only have access to demonstrations, $\mathcal{M}$, a dataset of $N$ collaborative task demonstrations $\mathcal{M} = \{\tau_i\}_{i=1}^N$. Each demonstration is a trajectory $\tau_i = (\boldsymbol{s}_0^i, \boldsymbol{a}_0^{E_1\,i}, \boldsymbol{a}_0^{E_2\,i}, \boldsymbol{s}_1^i, \boldsymbol{a}_1^{E_1\,i}, \boldsymbol{a}_1^{E_2\,i}, ..., \boldsymbol{s}_{T_i}^i)$. The goal of MA-GAIL [26] is to train the co-policy $\pi_{co}$ to imitate expert policy $(\pi_{E_1}, \pi_{E_2})$ by minimizing distance between the generated state-action distribution $\rho(\pi_{co})$ and the expert's distribution $\rho(\pi_{E_1}, \pi_{E_2})$ measured by Jensen-Shannon divergence. We optimize the following objective:

$$\min_{\pi_{co}} \max_{D} \mathbb{E}_{\boldsymbol{x} \sim \rho(\pi_{E_1}, \pi_{E_2})}[\log D(\boldsymbol{x})] + \mathbb{E}_{\boldsymbol{y} \sim \rho(\pi_{co})}[\log(1 - D(\boldsymbol{y}))], \tag{1}$$

where $D$ is a discriminative classifier which tries to distinguish state-action pairs from the trajectories generated by $\pi_{co}$ and $(\pi_{E_1}, \pi_{E_2})$.

However, directly applying MA-GAIL and equation 5 results in a poor co-policy due to the diverse behaviors in human-human demonstration. Given the same environment state $s$, human might perform different types of action $a^H$ based on their choice of strategy, which is hidden from the observable state. To disentangle the diverse human behaviors associated with the same state, we include a latent representation, $\boldsymbol{z}$, called *strategy*, to the input of the co-policy, $\pi_{co}(\boldsymbol{a}^H, \boldsymbol{a}^R | \boldsymbol{s}, \boldsymbol{z})$. We assume that the strategy can be inferred from the history of observations and the the human action and define a recognition model $\boldsymbol{z}_t \sim \psi(\cdot | \boldsymbol{s}_{t-K:t}, \boldsymbol{a}_{t-1}^H)$, where $K$ is the length of the history. To simplify the notation, we denote the history at time step $t$ as $\boldsymbol{h}_t = (\boldsymbol{s}_{t-K:t}, \boldsymbol{a}_{t-1}^H)$. The recognition model $\psi$ is trained jointly with the co-policy (Figure 2a). Our framework allows the strategy to evolve as the task progresses in an interaction episode. As such, the robot must constantly update its estimate of the human's latent strategy using $\psi(\cdot)$.

## 3.3 Uncovering diverse human behaviors

Introducing the latent strategy alone does not lead to successful co-policy learning. One challenge is that the human policy can simply ignore the latent strategy $\boldsymbol{z}$ and always opt to generate the most common behaviors. To this end, we use an information-theoretic regularization [45, 25] to enforce high mutual information between the latent strategy and the human action (Figure 2a). Given a prior distribution of the strategy space $p(\boldsymbol{z})$, we minimize the objective:

$$L_z = \mathbb{E}_{\boldsymbol{z} \sim p(\boldsymbol{z}), \boldsymbol{h} \sim \rho(\pi_{co}^H(\cdot, \boldsymbol{z}))} ||\psi(\boldsymbol{h}) - \boldsymbol{z}||, \tag{2}$$

where $\boldsymbol{h}$ is the length-$K$ trajectory generated from the acting policy $\pi_{co}^H$ given a sampled code $\boldsymbol{z}$. While $L_z$ indeed incentivizes the human policy to utilize the latent strategy when producing actions, it does not address another issue due to the imbalance distribution of training data, which results in $\psi(\cdot)$ ignoring less frequently seen $\boldsymbol{a}^H$ and mapping them to the same strategies that reflects the majority of $\boldsymbol{a}^H$. Whereas in practice, the human collaborator may interact with the robot with any strategy included in the demonstration. To address this issue, we introduce another objective that measures the reconstruction error of human actions in the training dataset (Figure 2b):

$$L_a = \mathbb{E}_{(\boldsymbol{h}_t, \boldsymbol{s}_t, \boldsymbol{a}_t^H) \sim p_{\mathcal{M}}} ||\pi_{co}^H(\boldsymbol{s}_t, \psi(\boldsymbol{h}_t)) - \boldsymbol{a}_t^H||. \tag{3}$$

In each training iteration, a batch of expert trajectory snippet $\boldsymbol{h}$ will be sampled from the dataset $\mathcal{M}$. The recognition model $\psi(\cdot)$ first estimates the hidden strategy $\hat{\boldsymbol{z}}$ from $\boldsymbol{h}$. The human policy

$\pi_{co}^H(\boldsymbol{s}, \hat{\boldsymbol{z}})$ is used as a decoder to reconstruct the expert action from the predicted strategy $\hat{\boldsymbol{z}}$. This loss encourages $\psi(\cdot)$ to encode different $\boldsymbol{a}^H$ to different $\boldsymbol{z}$ in order to minimize the reconstruction error, enforcing the inverse mapping from the human behavior space to the strategy space. Putting it together, the final objective for learning diverse collaboration behaviors becomes:

$$\min_{\psi, \pi_{co}} \max_{D} \mathbb{E}_{\boldsymbol{x} \sim p_{\mathcal{M}}}[\log D(\boldsymbol{x})] + \mathbb{E}_{\boldsymbol{y} \sim \rho(\pi_{co})}[\log(1 - D(\boldsymbol{y}))] + \lambda_1 L_z + \lambda_2 L_a, \qquad (4)$$

where $\lambda_1, \lambda_2$ are the hyperparameters for the intention and expert behavior reconstruction regularization term. The overview of the proposed algorithm is shown in Algorithm 1.

---

**Algorithm 1:** Co-GAIL - Learning diverse collaboration behavior

---

**Data:** $\mathcal{M} \leftarrow$ given human-human collaboration dataset
**Input:** $\pi_{co} \leftarrow$ initialize co-policy, $\psi \leftarrow$ initialize strategy recognition network, $D \leftarrow$ initialize the discriminator, $\mathcal{B} \leftarrow \emptyset$ initialize the replay buffer

1 **while** *not done* **do**
2      **for** *trajectory i=1, . . . , m* **do**
3          $\boldsymbol{z}_i \sim p(\boldsymbol{z})$ sample strategy code from prior
4          $\mathcal{B} \leftarrow \mathcal{B} \cup \{(\boldsymbol{s}_t, \boldsymbol{a}_t^H, \boldsymbol{a}_t^R, \boldsymbol{z}_i)\}_{t=1}^T$: append experiences from $\pi_{co}(\cdot, \boldsymbol{z}_i)$ to the buffer
5      **for** *update step = 1, . . . , n* **do**
6          $b^{\mathcal{M}}, b^{\mathcal{B}} \leftarrow$ sample batch of trajectories from $\mathcal{M}$ and $\mathcal{B}$
7          $L_z \leftarrow$ calculate strategy reconstruction loss with $\psi$ and $\pi_{co}$ (Eq. 2) with data from $b^{\mathcal{B}}$
8          $L_a \leftarrow$ calculate behavior reconstruction loss with $\psi$ and $\pi_{co}$ (Eq. 3) with data from $b^{\mathcal{M}}$
9          update $\psi$ and $\pi_{co}$ with the gradient of $\lambda_1 L_z + \lambda_2 L_a$
10          update discriminator $D$ according to Eq. 4 using $b^{\mathcal{M}}$ and $b^{\mathcal{B}}$
11      update co-policy $\pi_{co}$ with PPO[46]

---

### 3.4 Human-in-the-loop task execution

During test time, we discard the human policy and only use the robot policy to control the robot who interacts with a real human user (Figure 2c). Different from prior work [25] where the latent code reconstruction network is only used during the training process for mutual information maximization, our learned recognition model $\psi(\cdot)$ is an important component for the robot policy to recognize its human collaborators' strategy and anticipate their future behaviors in real-time. At each time step, $\psi(\cdot)$ takes a history of states $\boldsymbol{s}$ and human's previous action $\boldsymbol{a}^H$ as input and outputs a strategy estimation $\hat{\boldsymbol{z}}$. The robot policy will then generate the robot action, $\boldsymbol{a}^R \sim \pi_{co}^R(\boldsymbol{s}, \hat{\boldsymbol{z}})$.

## 4 Experiment Setup

The major advantage of **Co-GAIL** is its capability of uncovering diverse collaboration behavior from the given demonstrations. In the experiment section, we aim to compare **Co-GAIL** with state-of-the-art methods for learning collaboration skills. We focus on whether each method can successfully complete the task and successfully learn the diverse collaboration strategies. We also invite four real users to interact with the trained robot policy and evaluate the performance for each method.

### 4.1 Baselines

We start by comparing with imitation learning designed for a single agent (i.e., treat human as part of the environment) and build towards the proposed method progressively.

**BC-single:** Behavior-cloning algorithm that only learns a robot policy. During training, human behaviors are regarded as a part of the environment and played back in an open-loop fashion.

**BC-GAIL:** Train the human and robot policy in isolation. First learn human policy with BC. Then use GAIL algorithm[24] to learn the robot policy by interacting with the learned human policy.

**MA-GAIL:** Implementation of MA-GAIL [26]. Both human and robot policies are learned with a shared GAIL objective (Equation 5) during the training process.

**MA-InfoGAIL:** Add the latent strategy code learning objective based on InfoGAIL[25] (Equation 2) as an additional learning objective to MA-GAIL.

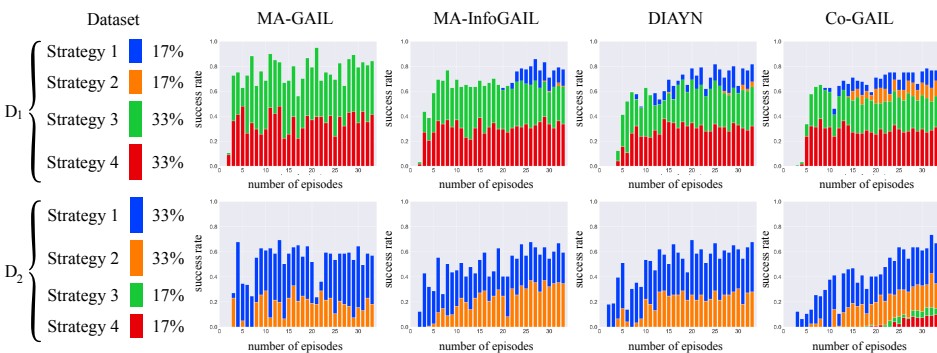

Figure 3: **Interpolation evaluation on 2D-Fetch-Quest for model trained with two differently unbalanced dataset.** Plots show the proportion of strategies generated by the model. Co-GAIL outnumbers the recovery of the two less represented strategies in all cases.

**DIAYN:** Implementation of [32] in a collaboration setting. Use a diversity reward to encourage the human policy to be diverse as possible.

**Co-GAIL:** The complete implementation of our proposed method. Our method differs from MA-InfoGAIL in the inclusion of behavior reconstruction loss (Equation 3) to encourage the model to cover all strategies in the given demonstration.

## 4.2 Collaborative tasks

We evaluate the methods in three continuous-action space domains that involve collaboration, illustrated in Figure 1. See Appendix for more details.

**2D-Fetch-Quest (2D-FetchQ.):** A collaborative 2D game inspired by the Fetch Quest minigame in Super Mario Party. Two agents must coordinate their strategy to fetch treasures from two locked rooms. Agents have complementary roles with temporal dependencies: one agent presses a button to unlock the room while the other agent fetches the treasure, allowing four different strategies depending on the order of the actions and roles. The human-human demonstrations exhibit four distinctive strategies. Strategy 1 and 2 differs in which party takes the treasure first. Strategy 3 and 4 are when the two parties decide to switch to the other side of the space to complete the task.

**HR-Handover (HR-H.):** A human-to-robot object handover task, in which the human holds an object and hands it over to the robot. The handover can take place in any region between the two agents with any approaching angle at any speed. The action space of each agent is the 6D position of the agent's hand relative to the previous time step and a float value describing the gripper state. The human-human demonstrations exhibit diverse handover locations, varying in all three dimensions. We visualized the distribution of the position when handover happens (X-Y locations) in Figure 4a.

**HR-SeqManip (HR-S.):** Place an object in a drawer by collaboratively manipulating both the object and the drawer. There are two possible strategies to solve this task: the robot assists by opening the drawer while the human drops the object, or the human hands over the object to the robot and opens the drawer, while the robot drops the object into the drawer. Other forms of diversity include the object handover locations and details in motion such as ways to open the drawer. The action space is the same as in HR-Handover.

These domains are implemented in an interactive simulation environment in which the human users control the agents via a control interface. Demonstrations in **2D-Fetch-Quest** are collected through a joystick interface. **HR-Handover** and **HR-SeqManip** are implemented in iGibson environments [16] with Bullet physics engine [15]. Collaborative demonstrations are collected from two human users controlling the two agents using the Roboturk teleoperation interface [47, 48]. More details of the environment design could be found in Appendix.

## 4.3 Experimental setup

We assess the performance of $\pi_R$ in terms of task completion (success rate) and coverage of diverse strategies. We employ three evaluation protocols to show different facets of the problem of learning diverse and plausible collaborative behaviors.

**Interpolation:** Uniformly sample the learned strategy space and simulate a robot-human trajectory for each strategy using the learned co-policy. We evaluate the success rate and diversity of these trajectories for our method and the baselines. This evaluation provides an insight into the effect of each strategy on the co-policy, and the coverage of diversity presented in the training data.

**Replay:** We subject the robot policy to interact with the *open-loop replays* of human trajectories previously unseen during training. Unlike interpolation evaluation, the replay evaluation tests of the policy can correctly respond to diverse and previously unseen collaborative behaviors. The setting allows scalable evaluation as we do not need a real human in the loop, but it is limited because the human policy does not react reciprocally to the robot.

**Real-human evaluation:** Proof-of-concept evaluation with task execution with a real human user in-the-loop interacting with the robot's policy as illustrated in Figure 2c. We report the success rate for all domains and methods.

## 5 Results

**Modeling human behavior improves $\pi_{co}^R$:** The results show that modeling human behavior improves the robustness of the learned $\pi^R$ when facing unseen human behavior. Our proposed Co-GAIL outperforms BC-single in replay evaluation in Table 1 for more than $20\%$ in all three tasks.

**Co-optimization outperforms learning in isolation:** Comparing the replay evaluation results (Table 1) for Co-GAIL and MA-GAIL indicates that co-optimizing the $\pi_{co}$ has benefits over the sequential training of human and robot policy in BC-GAIL.

Table 1: **Replay evaluation:** Success rate (mean and 95% confidence interval over three seeds).

|  | Replay | | |
|---|---|---|---|
|  | 2D-FetchQ. | HR-H. | HR-S. |
| **BC-single** | $21.1 \pm 1.9$ | $13.3 \pm 3.3$ | $15.0 \pm 2.3$ |
| **BC-GAIL** | $27.2 \pm 5.0$ | $14.1 \pm 4.6$ | $16.6 \pm 7.1$ |
| **MA-GAIL**[26] | $30.0 \pm 3.4$ | $23.7 \pm 5.3$ | $21.0 \pm 3.7$ |
| **MA-InfoGAIL**[25] | $40.0 \pm 1.7$ | $27.8 \pm 4.4$ | $26.6 \pm 8.8$ |
| **DIAYN**[32] | $44.4 \pm 8.4$ | $22.7 \pm 1.5$ | $18.3 \pm 7.1$ |
| **Co-GAIL (ours)** | $\mathbf{53.3 \pm 4.4}$ | $\mathbf{43.9 \pm 6.3}$ | $\mathbf{40.0 \pm 8.2}$ |

**Co-GAIL learns more diverse strategies compared with the baselines:** Unlike existing methods, we designed Co-GAIL to obtain a $\pi_{co}^R$ that can interact in the scenario that a real human approaches the task with diverse strategies. Using the 2D-Fetch-Quest domain, which can be solved with 4 distinct strategies (see Appendix), we designed two datasets containing different distribution of demonstrated strategies as follows: $D_1 = \{17\%, 17\%, 33\%, 33\%\}$ and $D_2 = \{33\%, 33\%, 17\%, 17\%\}$, as illustrated in Figure 3.

We train relevant baselines over each dataset and use the interpolation evaluation to analyze the distribution of learned strategies in the latent space. We observe that Co-GAIL is able to recover all the strategies, while the baselines are not able to generate the less represented strategies or have comparatively fewer of them. Similar results are also observed in the high-dimensional task HR-Handover.

Table 2: **Interpolation evaluation:** Success rate (mean and 95% confidence interval over three seeds).

|  | Interpolation | | |
|---|---|---|---|
|  | 2D-FetchQ. | HR-H. | HR-S. |
| **MA-GAIL**[26] | $78.5 \pm 7.2$ | $99.2 \pm 1.4$ | $46.2 \pm 10.3$ |
| **MA-InfoGAIL**[25] | $80.2 \pm 3.0$ | $96.4 \pm 4.1$ | $51.7 \pm 7.5$ |
| **DIAYN**[32] | $80.2 \pm 3.6$ | $86.2 \pm 9.0$ | $45.4 \pm 7.9$ |
| **Co-GAIL (ours)** | $82.1 \pm 3.1$ | $81.2 \pm 3.1$ | $31.6 \pm 2.5$ |

We analyze the range of 3D locations in which the model interpolation performs a successful human-to-robot handover (HR-Handover). A top view of the space between the human and the robot is shown in Figure 4a for two different random seeds. Handover locations of MA-InfoGAIL tend to concentrate instead of covering the demonstrated distribution. We also observe that, since DIAYN encourages the learned policy as diverse as possible, it also explores strategies that are less plausible in real human-robot collaboration. Co-GAIL (green dots) shows better coverage across the range of the dataset on both the X and Y dimensions (Z in Appendix due to space). We further investigate the interpolation results of each method and find that MA-GAIL, MA-InfoGAIL, DIAYN have high success rates during interpolation (Table 2) while performing poorly in replay evaluation (Table 1), indicating that previous methods tend to overfit, while Co-GAIL explores the dataset more broadly.

**The learned latent space maps to qualitatively different human strategies (Figure 4b):** In this experiment, we verify if regions of the learned latent space correspond to distinct recognizable human strategies. For HR-Handover, we divide the handover locations projected on the horizontal

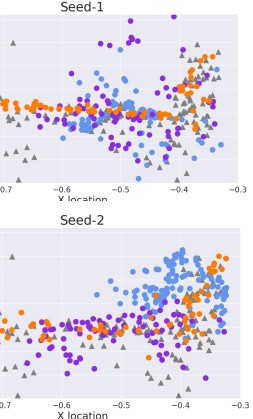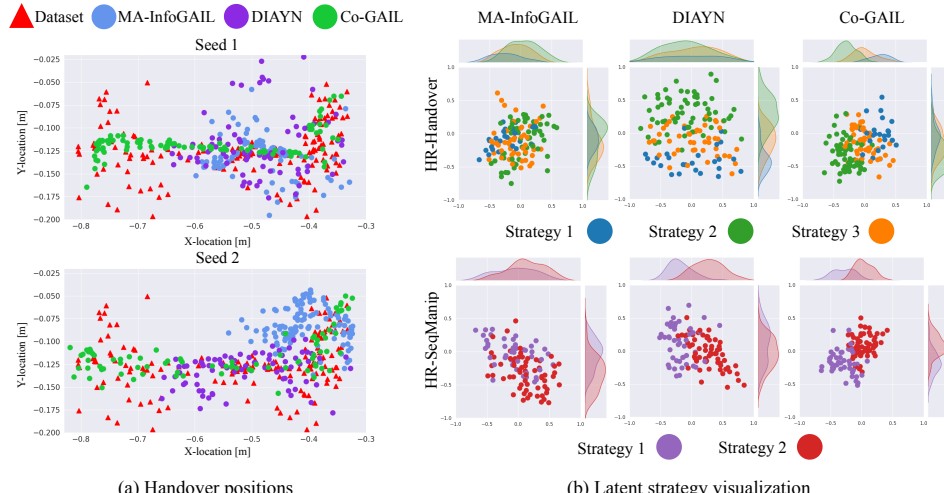

(a) Handover positions       (b) Latent strategy visualization

Figure 4: **Diversity and strategies encoded in latent space**. (a) Top view of successful handover locations from HR-Handover interpolation. (b) Visualization of the correspondence of learned latent strategy space (2D domain) to different type of manipulation strategies (colors).

plane XY into three regions, obtaining handovers executed at the left ($S_1$), middle ($S_2$), and right ($S_3$) sides. For HR-SeqManip, we determined two possible strategies ($S_1, S_2$) depending on the human and robot roles (see Appendix). We use the recognition model of each method to compute an average latent code for each unseen trajectory and visualize the code in Figure 4b, where each code is colored by its strategy category. Codes for the same strategy tend to cluster in the same region. We observe a better separation of latent strategy learned using Co-GAIL than the other two methods. This shows the value of our diversity objective in Equation 3.

**Real human user interacting with** $\pi_{co}^{R}$: We performed a proof-of-concept analysis with real users to assess representative methods. Users interacted with all methods in randomized order. We report the success rate out of 20 trials per user per method per domain in Table 3. Co-GAIL consistently outper-

Table 3: **Real-human evaluation**

| User id | 2D-FetchQ. 1 | 2 | 3 | 4 | HR-H. 1 | 2 | 3 | 4 | HR-S. 1 | 2 | 3 | 4 |
|---|---|---|---|---|---|---|---|---|---|---|---|---|
| **BC-single** | 70 | 65 | 50 | 65 | 20 | 65 | 10 | 20 | 10 | 5 | 10 | 15 |
| **MA-GAIL**[24] | 65 | 50 | 45 | 50 | 50 | 35 | 25 | 20 | 30 | 25 | 35 | 25 |
| **MA-InfoGAIL**[25] | 85 | 80 | 40 | 50 | 55 | **90** | 65 | 25 | 35 | 40 | 40 | 50 |
| **Co-GAIL** | **100** | **90** | **85** | **80** | **75** | 70 | **70** | **75** | **60** | **50** | **65** | **70** |

forms the other methods except for one case in the HR-Handover. In this case, the user performed handover in a small area that overlapped with the region that MA-InfoGAIL overfit to. Co-GAIL showed the best adaptation to diverse strategies across users and trials. While a small sample size, these promising results are the first demonstration of a learning method that succeeds at adapting to real users in these challenging tasks with continuous motion and varied strategies.

## 6 Conclusions and Future Work

We present Co-GAIL, a method for training robot assistants that can handle diverse human behaviors. Our method represents a novel approach for taking advantage of simulated environments in HRI research. This is achieved by leveraging data of human-human collaboration demonstrations as guidance to concurrently generate simulated interactive behaviors and train a human-robot collaborative policy. We test Co-GAIL in a 2D game and two high-dimensional manipulation tasks, with both procedural evaluation and online human-in-the-loop evaluation. While Co-GAIL outperforms other state-of-the-art methods in challenging HRC tasks, a few challenges remain open for future works. In our current implementation, the roles of leaders and followers must be assigned to the human agent and the robot agent in advance. Enabling the agents to flexibly switch roles online would further improve the collaboration experience. Another future direction is to apply the learned policy to a real robot. A possible approach is to align the observation space of the simulated environment and the real world through state estimation and employ domain randomization [49] to allow the agent to be robust against domain shift in sim-to-real transfer.

**Acknowledgments**

We thank the Stanford Institute for Human-Centered Artificial Intelligence and the HAI-AWS Cloud Credits for Research program for its support with compute resources. Chen Wang thanks Lihua Wang and Zhongqing Zheng for helping with the development of the data collection system.

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
