# OpenReview forum: "Co-GAIL: Learning Diverse Strategies for Human-Robot Collaboration"
_robot-learning.org/CoRL/2021/Conference — CoRL2021 Poster_

### Official Review · Reviewer_wpop · 2021-07-06

**Originality:** Good
**Technical Quality:** Fair
**Clarity Of Presentation:** Excellent
**Impact:** 3

**Recommendation:**

Weak Accept: I recommend accepting the paper, but will not argue for my recommendation if the majority of other reviewers have a different opinion.

**Summary:**

The authors made a good faith attempt to improve the quality of their evidence, and to improve the means by which they analyzed the evidence. As a result of this, I am more confident that Co-GAIL is better than their baselines, although I am still somewhat confused by the evidence. In particular, a T3 Dunnett post-hoc test found that Co-GAIL was only significantly superior to MA-InfoGAIL in one of three test environments, and was not significantly superior in another two. However, a GLM predicting success rate against the model, task, and user found that the weight on Co-GAIL was significantly higher than the weights on the other method terms. The only differences between these comparisons is that the GLM aggregates across tasks, and has a per-user term (which seems justified with n=20 trials per user). I am not an expert statistician, but these results taken together seem to say that Co-GAIL is a lot better on some tasks, and not obviously worse on the rest. The authors also gave a strong explanation for why my interpretation of Table 2 was incorrect, and the explanation in their paper was likely to be correct. This is enough for me to shift my judgement from weak reject to weak accept. The original review follows unchanged.

===

Doing imitation learning in an environment that requires interaction with a human is not handled well by most imitation learning methods, as they tend to assume any human involvement is in the form of a fixed human policy that does not react to the robot. This is either false (if the robot's state is not well captured in the state) or difficult to model and collect data on (when it is).

In order to address this issue, there exist a collection of methods that attempt to estimate both a human and a robot policy from a collection of demonstrations where humans act as both the human in the environment and the robot helping them accomplish a task. The most successful of these methods are MA-GAIL, which adapts the GAIL framework to a multi-agent setting, MA-InfoGAIL, a modification of the above that considers a latent strategy variable, and DYIAN, which tries to maximize a diversity criteria. MA-InfoGAIL and DYIAN have policies that are conditioned on a strategy latent variable that attempts to capture the fact that humans will act according to different strategies even in the same state.

The novel contribution of this paper is to introduce Co-GAIL, a modification of MA-InfoGAIL that includes an additional loss designed to prevent mode collapse in the estimation of the latent strategy variable. This new method (and several baselines) are evaluated on a 2D-gridworld and two 3D physics based object manipulation tasks. The authors find that, according to some evaluations using historical (fixed) human data and a small pilot study of humans reacting live to the robot's actions, co-GAIL outperforms all of the baselines overall. They also prevent evidence that mode collapse is a problem in prior methods.

**Issues:**

My biggest issue is that three replications simply isn't enough to be confident that your method is actually superior to the baselines. In 2D-FetchQ the 95% confidence intervals (assuming a Gaussian, which is hard to do when n=3) of DYIAN and Co-GAIL overlap. This is not the case in the 3D environments, which is something, but the variances of your method in those environments is also quite high, especially for n=3. It's also not reassuring that your method did worse than the others in the interpolation setting. It is true that the other methods that learn mode collapsed policies have an easier task here, but it betrays a weakness in your policy learning step if you can't learn a robot policy that is near optimal under your own model of the human.

Ultimately both of the simulated human cases provide some evidence, but aren't conclusive. Replay is limited in that the human can't react to the robot's actions, which was a major part of the motivation for this paper. Interpolation does have a reactive human, but lacks external validity. The human pilot study is the best evidence available, but unfortunately there were only four subjects available. At least on the 2D environment, you can get subjects over the internet. It is hard to view four replications as strong evidence. Admittedly the scatterplots for 2D-FetchQ and HR-S look fairly separate, but there is definitely room for DYIAN to come out on top in the handover task with more data. I would also recommend you produce those scatterplots, and also report mean scores across different subjects, at least in the appendix.

I also have a conceptual issue with how the interpolation experiments were done in terms of the reported success rate. When training, you learn a psi function to map human actions to a latent strategy. An ideal psi function captures multiple human modes of behavior, even at identical states. A psi that is spread out over all the different values in the strategy space is not very informative for the policy. So we expect that the empirical values of z will concentrate around a few different modes. If the goal of the reported success rate is to give a self-consistency check on the co-policies, taking a uniform prior over strategies seems inappropriate. What I would do in your position would be to record the z values along the fixed human trajectories you have, visually (or automatically) identify a few mode points for each method, and then use those mode strategy values to simulate the human's policy. Using a uniform prior to identify the diversity of human trajectories and compare that to the prerecorded trajectories is reasonable.

As an extremely minor issue, some implementations of GAIL use log(D(.)) as the proxy reward, and some use -log(1-D(.)). For the sake of completeness it would be nice if you could mention this somewhere in the appendix.

There are several grammatical issues, notational inconsistencies, and inconsistencies between the appendix and the main paper. I expect all of these can be fixed quite simply, and I provide them mainly for the benefit of the authors.

62: "agents are considered" -> "agents are assumed"
84: "which is crucial especially" -> "which is especially crucial" or "which is crucial, especially"
112, 126: $\mathcal{M}$ is both the Markov game and the set of demonstrations, pick a different name for one or the other
131: "that the start state is $s_i \sim \rho_0(\cdot)$" -> The way this is written was confusing, and this is such a standard assumption that I don't think you need to mention it at all.
143: "$\mathbf{z}_t \sim \psi(\mathbf{z}_t | ...)$" -> ""$\mathbf{z}_t \sim \psi( \cdot | ...)$"
155: "when producing action" -> "when producing actions"
191: "Equation1" -> "Equation 1" You're missing a space.
192: "infoGAIL" -> "InfoGAIL" Capitalization.
212: "collaboratively manipulate" -> "collaboratively manipulating" or rework the sentence more broadly.
245, 249, 277: You appear to have flipped table 1 and table 2 in several places.
298: "MA-InfoGAIL overfits to" -> "MA-InfoGAIL overfit to" tense issues.
315: "to the real robot" -> "to a real robot"

Table 1, 2, appendix: The main paper says you report variances, but the appendix says you report standard deviation. As a notational issue, X +- Y is only for something which is an interval, like a confidence interval. If you want to report a mean and variance or a mean and standard deviation, the notation is X (Y). Additionally, if you're going to bold specific items that are the best, you need to bold all items that COULD be the best, meaning that you bold the item with the best mean, and all other rows whose confidence intervals overlap with the confidence interval of the row with the best mean.

**Reviewer Expertise:**

Good: General knowledge of the area

**Strengths And Weaknesses:**

Strengths:
The paper has a very strong introduction, and a very clear explanation of their method. I was not familiar with the MA-GAIL line of work, but was able to follow their paper.

The novel contribution is a natural solution to the problem at hand.

The paper provides evidence that the problem it is trying to fix actually exists, which is something many neglect to do.

The paper compares its method to a fairly comprehensive set of baselines.

Weaknesses:
The quality of evidence is low in many experiments. Only 3 random seeds were used in the simulated human experiments, and the reported variances are quite high. Only one replication was performed on the human data, which is understandable for a pilot in present circumstances, but it makes it hard to strongly believe your results.

If I understand things correctly, the interpolation experiments have a significant conceptual flaw. Considering a uniform prior over strategy variables seems very inappropriate when we expect the strategy encoder to eventually concentrate around a few modes.

**Summary Of Recommendation:**

I cannot recommend this paper until further replications are performed. The simulated experiments only use three replications, and the human trials only use four subjects each. The spread in data values is very often large enough to be concerning. The method itself is reasonable, and qualitatively CoGAIL does appear to match the human's trajectory distribution better, but I simply cannot be confident that the method actually outperforms DYIAN in all cases.

---

> ### Author Response · Authors · 2021-08-23
> **Response to Reviewer wpop**
>
> Thanks for your thorough and helpful comments! We are glad to know that you agreed with our contribution of solving the diversity problems for human-robot collaboration, which are commonly neglected by prior works. You also mentioned our comprehensive comparison with baselines provides strong evidence to support the proposed method. You brought up some great questions and suggestions. Please find our answers below:
>
> - **Only 3 random seeds were used in the experiments:** We conducted additional two runs with new seeds for Co-GAIL and two baseline methods in the 2D Fetch-Quest and HR-Handover environment (HR-SeqManip will be added later due to the time limit). Please find attached is the new replay evaluation result (mean and 95% confidence interval over 5 seeds.) The advantage of Co-GAIL over the baselines is consistent: the mean success rate of Co-GAIL is 8% above DIAYN in 2D Fetch-Quest and more than 20% in HR-Handover. For more details of why Co-GAIL is better than DIAYN, please refer to our reply to the fourth question.
> Methods|2D FetchQ. (3 seeds)|HR-H. (3 seeds)| 2D FetchQ. (5 seeds)| HR-H. (5 seeds)
> :-|:-:|:-:|:-:|:-:
> MA-InfoGAIL | 40.0 ± 1.7 | 27.8 ± 4.4 | 38.6 ± 6.8 | 26.6 ± 7.7
> DIAYN |  44.4 ± 8.4 | 22.7 ± 1.5  | 42.9 ± 8.2 | 20.7 ± 0.9
> Co-GAIL| 53.3 ± 4.4 | 43.9 ± 6.3 | 51.2 ± 5.5 | 44.2 ± 5.4
>
> - **Only 4 subjects were available in the real human evaluation:** We invited another two participants into our pilot study. Please find the following additional results for participant 5 and 6, and mean and standard deviation over 6 participants. Co-GAIL achieved the best performance in all three environments. We also observed that the results of MA-GAIL and MA-InfoGAIL have large std. This is because these methods tend to focus on learning a limited set of familiar strategies and ignore the more challenging cases. As a result, the success rates vary widely depending on whether the participants prefer the more challenging strategies (from the policy’s perspective) or the strategies that the policy excels at. The observation also indirectly supports our finding in the interpolation evaluation, which is discussed in the next question. In contrast, Co-GAIL has a relatively small std in all three environments because it has successfully captured diverse strategies during the training process.
> |  |  |  | 2D-Fetch-Quest |  |  | HR-Handover |  |  |HR-SeqManip |
> |---|:---:|:---:|:---:|:---:|:---:|:---:|:---:|:---:|:---:|
> | User id | 5 | 6 | Mean(std) | 5 | 6 | Mean(std) | 5 | 6 | Mean(std) |
> | MA-GAIL | 20 | 45 | 45.8(13.4) | 20 | 30 | 30.0(10.4) | 20 | 15 | 25.0(6.5) |
> | MA-InfoGAIL | 20 | 55 | 55.4(22.4) | 50 | 25 | 50.8(23.7) | 35 | 35 | 39.2(5.3) |
> | Co-GAIL | 85 | 80 | 86.7(6.9) | 75 | 65 | 71.7(3.7) | 50 | 60 | 59.2(7.3) |
>
> - **Uniform prior is inappropriate for interpolation evaluation:** The interpolation experiment offers insights into an algorithm’s ability to learn diverse strategies from the given dataset. Visualizing key measurements such as strategy categories (Fig. 3) and handover locations (Fig. 4a) by uniformly sweeping over the strategy code space allows direct qualitative comparisons among the strategy diversity captured by different algorithms. What we observed in Figure 3 and 4(a) is that the baseline methods tend to focus on generating a limited set of familiar strategies and ignore the more challenging cases with less training data coverage. Furthermore, the success rate metric reported in Table 2 indirectly supports the observation. The baseline methods receive higher success rates in interpolation evaluation than Co-GAIL. This is because mapping the entire code space to a subset of the strategy distribution allows the policy model to focus its learning capacity to the few strategies with more training data. In contrast, Co-GAIL generates more diverse interactive behaviors, but at the cost of subjecting the policy to more challenging / less familiar strategies. Finally, uniformly sweeping the learned latent space to evaluate diversity of generated samples is widely used in prior generative learning methods such as infoGAIL[25] and infoGAN[45] and image generation[Karras et al. StyleGAN]. While we agree that finding and leveraging dominating modes in human behaviors through post-hoc analysis may further differentiate the characteristics of different algorithms, we fear that adding such heuristics-based elements (e.g., ways to select modes) may introduce bias to the evaluation process and possibly cause unfair comparison.

---

> > ### Author Response · Authors · 2021-08-23
> > **Response (Part 2)**
> >
> > Please note that our response has been split into two parts due to space constraints. (Part 2/2)
> >
> > - **Co-GAIL vs DIAYN:** We observed 20% performance boost in two manipulation tasks and 10% in the 2D Fetch-Quest task when comparing Co-GAIL with DIAYN. One of the critical technical issues of DIAYN is its learning objective that encourages exploring strategies "as diverse as possible"[32], which encourages the algorithm to search diversity in an unbounded manner among all dimensions of the strategy. Such incentive encourages the policy to venture out of the distribution of realistic human behaviors. For example, in Fig.4a, DIAYN (purple dots) failed to discover the left region (x<-0.6) and keep exploring the top area(y>-0.075) that is beyond reasonable human motions. In contrast, CoGAIL (green dots) covers most parts of the main distribution. The issue is exacerbated in larger environments, as higher state space dimension and size further incentivizes unbounded exploration. We observe that compared to the 2D Fetch-Quest domain, DIAYN suffers significant performance drops in larger 3D environments in HR-Handover and HR-SeqManip. In contrast, Co-GAIL successfully explores diversity within the range of reasonable human behaviors (red triangles in Fig.4a), which does not have this unbounded searching issue.
> >
> > - **GAIL formulation:** Thanks for pointing this out! We use $-log(1-D(x))$ as the reward function for the 2D Fetch-Quest task, and LSGAN[50] for the 3D manipulation tasks (HR-Handover, HR-SeqManip). More details are available in Appendix B.
> >
> > - **X+-Y policy:** We use the confidence interval of 95% to report results in the tables. We have fixed the typo in the Appendix.
> >
> > - **Grammatical issues:** Thanks! We have fixed them in the revision.

---

> > > ### Comment · Reviewer_wpop · 2021-08-23
> > > **Thanks**
> > >
> > > As above, I'm more confident that Co-GAIL is robustly better than DYIAN, but I would appreciate a proper statistical analysis.
> > >
> > > I appreciate all of the minor fixes!

---

> > ### Comment · Reviewer_wpop · 2021-08-23
> > **I appreciate the additional data**
> >
> > I appreciate the additional runs in simulation, and understand that running more replications take time. These results look much stronger, and if the trend continues when the HR-SeqManip results get back I will be very happy.
> >
> > I assume for the human study you meant for the columns to read "2D-FetchQ. HR-H. HR-S" as in the original paper, instead of listing 2D-Fetch-Q three times?
> >
> > The human study result for the experiment in the third column looks rock solid. The experiment in column 1 is somewhat borderline (unequal variance difference of means tests p=0.017 uncorrected for multiple comparisons, which goes just over 0.05 when you do a Bonferroni correction for the 3 choose 2=3 multiple comparisons being done), and the experiment in column 2 is a lot worse by the numbers, with an uncorrected p=0.083. The reason the second column experiments are so inconclusive is in part because DYIAN has such a high variance, which itself is undesirable.
> >
> > All of the above stats are quick and dirty. I would recommend you do a more principled ANOVA post-hoc analysis in order to get the best results out of your low n. https://www.ijntse.com/upload/1447070311130.pdf recommends a Dunnett T3 test for low-n unequal variance data. I think the right way to do this analysis is to give people the statistics, but note that the p-value is only about comparing the means, and low variance is also inherently desirable. n=5/6 is just on the border of what statisticians consider enough data for doing an ANOVA, so it would probably strengthen the paper a lot if you could perform even more replications. Again, I understand there is only so much time.
> >
> > Figure 3a and 4 are useful and I have no problem with them. My issue is only with Table 2. The explanation you gave is consistent with the data, and it seems likely to be true. I'm not sure it's *evidence* of anything about Co-GAIL, because I could easily imagine a story for why Co-GAIL should be better. Consider:
> > "DYIAN and the MA-GAIL family of methods experience mode collapse in the strategy encoder, and so the strategy conditional policies they learn become overfit to a constant strategy, and thus do very poorly on all of the other strategies, which uniform sampling penalizes. Because Co-GAIL's policy has to be good at multiple strategies, it generalizes better across the full range of strategy values."
> > Basically, a-priori there were two ways that prior methods could have reacted to strategy mode collapse: a) the policy could have overfit to a near-constant strategy variable or b) it could have learned to ignore the strategy variable entirely. If a were true, I would expect Co-GAIL to do better than the baselines, and if b were true I would expect Co-GAIL to do worse than the baselines. Table 2 resolves this in favor of the latter, but this says more about the baselines than it does about Co-GAIL. If a vs b was already resolved in prior literature that I'm not aware of, then Table 2 says more about Co-GAIL, but I didn't get that from the text.

---

> > > ### Author Response · Authors · 2021-08-25
> > > **Additional Discussion for Table 2**
> > >
> > > We believe there are several points we could address to help understand the results in Table 2.
> > >
> > > - *“DYIAN and the MA-GAIL family of methods experience mode collapse in the strategy encoder, and so the strategy conditional policies they learn become overfit to a constant strategy, and thus do very poorly on all of the other strategies, which uniform sampling penalizes.”*
> > >
> > > Uniformly sampling the code space does not penalize overfitting in terms of success rate in the interpolation experiment (Table 2).  InfoGAIL-like methods establish mapping between the code and the strategy in an unsupervised manner, meaning that the algorithm is free to assign any code to any strategy, as long as the policy can achieve high rewards (i.e., fool the discriminator). One of the easiest ways to achieve high reward during training is simply mapping the entire code space to a small subset of strategies that are easy to accomplish. And this is what we refer to as “overfitting”. As a result, when we uniformly sample the code space in the interpolation experiments, methods that overfit would achieve high success rates because their policy model simply maps all the sampled codes to a (or a few) constant strategy. And because Co-GAIL encourages the robot policy to learn a diverse set of strategies, it underperforms DIAYN and MA-GAIL in Table 2: the success rate metric in Table 2 counts episodes where the goal is accomplished, regardless of which strategy is used to accomplish it.
> > >
> > > - *“Basically, a-priori there were two ways that prior methods could have reacted to strategy mode collapse: a) the policy could have overfit to a near-constant strategy variable or b) it could have learned to ignore the strategy variable entirely.”*
> > >
> > > Case (a) should not happen because of the nature of the learning setup: the unsupervised learning setup forces the policy to map the entire code space to strategies. During training, the strategy code input is randomly sampled within a range (two floats in range [-1.0, 1.0] in our settings).The policy needs to interpret each sampled code and map it to a behavior that can potentially fool the discriminator. Situation (b) is incorrect too, since neglecting the code will cause the policies to always output exactly the same trajectory (assuming environment dynamics is deterministic) which will be easily discovered by the discriminator and cause the model to receive low rewards. The overfitting phenomenon is closer to (b) than (a). But rather than totally ignoring the code, the baseline methods learn to map the entire code space to a few strategies (as shown in Fig. 3a and Fig. 4), with variations in low-level movement trajectories just so that they can fool the discriminator to get reward.
> > >
> > > - *“I could easily imagine a story for why Co-GAIL should be better.”*
> > >
> > > We believe that results reported in Table 2 highlights the key difference between the Co-GAIL and the baselines. The interpolation success rate metric reported in Table 2 is an indirect support for the observation we found in the interpolation visualization results in Figure 3 and 4(a), where the baseline methods tend to focus on generating a limited set of familiar strategies and ignore the more challenging cases with less training data coverage. DIAYN and MA-GAIL map the entire code space to a subset of the strategy distribution, which allows the policy model to focus its learning capacity on the few strategies with more training data. In contrast, Co-GAIL generates more diverse interactive behaviors, but at the cost of subjecting the policy to more challenging / less familiar strategies.

---

> > > > ### Comment · Reviewer_wpop · 2021-08-30
> > > > **Table 2 discussion**
> > > >
> > > > Thank you for this explanation. I found bullet 2 to be particularly helpful. I now agree that Table 2 means basically what you claim in means in the paper.

---

> > > ### Author Response · Authors · 2021-08-25
> > > **Post-hoc Analysis**
> > >
> > > We conducted a post-hoc analysis to assess the results of the evaluation with users, keeping in mind that our study is a pilot with n=6 (n=4 in original submission, and added 2 more users), given the difficulties of testing with real users in the current COVID-19 situation.
> > >
> > > We first conducted the test suggested by the reviewer, ANOVA post-hoc analysis with the T3 test of Dunnett (see results in table below). The success rate of MAGAIL vs. CoGAIL have a significant difference for the three domains (2D-Fetch-Quest, HR-Handover, HR-SeqManip), whereas MAInfoGAIL vs CoGAIL is trending toward significance for 2D-Fetch-Quest but not for HR-Handover and HR-SeqManip. These results, while from a low sample size, are a promising signal of how Co-GAIL improves performance over the baselines.
> > >
> > > | Dunnett T3 |  | 2D-Fetch-Quest |  | HR-Handover |  | HR-SeqManip |
> > > |---|:---:|:---:|:---:|:---:|:---:|:---:|
> > > |  | t value | Pr(>\|t\|) | t value | Pr(>\|t\|) | t value | Pr(>\|t\|) |
> > > | MAGAIL - CoGAIL | -6.078 | 0.001372 | -8.427 | 0.000364 | -7.833 | 0.000032 |
> > > | MAInfoGAIL - CoGAIL | -3.015 | 0.062354 | -1.947 | 0.26002 | -4.94 | 0.010105 |
> > >
> > > In addition to the reviewers suggestion, we analyze the significance of the comparisons by using a model that takes into account the difference in the performance baseline for each user. Furthermore, while our initial submission showed the results for success_rate only, here we have included the task_time (time it takes for the user to complete the task) for additional evidence of how Co-GAIL generates a better interactive assistant by reducing the task time vs. the baselines with significance.
> > >
> > > We fit a generalized linear mixed-effect model to test the main effects of the 'method' and 'domain' over the success rate (1) and the task time (2), with random effects of the subject_id to account for the natural differences in performance of each human user. We applied the Holm–Bonferroni correction. We fit this model in R using https://www.rdocumentation.org/packages/lme4/versions/1.1-27.1/topics/glmer
> > >
> > > (1) success_rate ~ method + domain + (1|subject_id)
> > >
> > > (2) task_time ~ method + domain + (1|subject_id)
> > >
> > > After fitting a model for each metric (success_rate, task_time), the results show a significant difference in performance between Co-GAIL and all the baselines (see tables in attached pdf -- p values and 95% confidence intervals). This result is also consistent for all domains.
> > >
> > > Additionally, we attach plots of the data for success_rate and task_time for visual inspection in the supplementary material. The full result is included in a standalone pdf (stats_analysis.pdf) in the supplementary material zip file.

---

> > > ### Author Response · Authors · 2021-08-25
> > > **Additional Seeds for SeqManip**
> > >
> > > Please find attached the HR-SeqManip results with 5 seeds. The trend indeed holds: Co-GAIL outperforms the baselines in all three domains.
> > >
> > > |  | HR-SeqManip (3 seeds) | HR-SeqManip (5 seeds) |
> > > |---|:---:|:---:|
> > > | MA-InfoGAIL | 26.6 ± 8.8 | 30.8 ± 4.3 |
> > > | DIAYN | 18.3 ± 7.1 | 20.0 ± 6.7 |
> > > | Co-GAIL | 40.0 ± 8.2 | 41.6 ± 6.0 |

---

### Official Review · Reviewer_RH4p · 2021-07-23

**Originality:** Good
**Technical Quality:** Good
**Clarity Of Presentation:** Very Good
**Impact:** 3

**Recommendation:**

Weak Accept: I recommend accepting the paper, but will not argue for my recommendation if the majority of other reviewers have a different opinion.

**Summary:**

This paper proposes a learning approach to train control policies that allow robotic agents to assist human partners in collaborative tasks.
The main insight is to incorporate the diverse range of behaviors/strategies that humans may follow during collaborative tasks into robot control policies for a more effective joint task execution.
Similar to prior work, the proposed approach first trains a collaborative policy using human-human demonstrations that results in a shared policy network that outputs actions for both agents (human and robot).
This policy learning is then extended, again similar to prior work, by integrating a latent representation of human strategies, i.e., by incentivizing the human policy to condition on the strategy to replicate demonstrated human actions.
To improve capturing diverse human behaviors, this latent representation and the human policy are also co-optimized as an autoencoder using demonstrations.
The results highlight comparable or better performance in terms of strategy coverage and responsiveness on the three tasks being evaluated.

**Issues:**

Please also see above (Strengths & Weaknesses).\
Some further improvement points:\
There is a strong emphasis on capturing diverse human behaviors, but even though this is supported by the interpolation evaluation on the 2D-Fetch-Quest task, looking at the other tasks' results (e.g., Fig4), I couldn't identify a significant (qualitative) difference between the proposed Co-GAIL approach and DYIAN.
Also, I'd recommend to not only focus on the positive results of the proposed approach but also where it underperforms, e.g., success rate of interpolation evaluations for the HRI experiments (Table 2).
I'd suggest using a better/fairer phrasing and/or quantitative comparison, especially to justify diversity claims.

Real human user interacting with robot policy: Even though it's a proof-of-concept, it's not clear what the protocol for this experiment was:
- who were the subjects? how much they had experience with robotics and/or this particular setup? are they PhD students, friends, random people? etc.
As a qualitative analysis, it might be fine to report on, but a quantitative analysis would be unreliable without a proper user study protocol.

small issues:
- typo: caption of Fig.1: 'Pick and object ...', line 212: manipulate -> manipulating, line 231: reply -> replay, line 271: '... instead (of) covering ...'
- misreference: line 152: Figure 2b -> 2a, line 160: Figure 2c -> 2b, line 245: Table 2 -> 1, line 277: Table 1 -> 2 and Table 2 -> 1


**Reviewer Expertise:**

Very good: Comprehensive knowledge of the area

**Strengths And Weaknesses:**

The main strength of this work is its learning architecture that effectively integrates human strategies.
This integration allows modeling human behaviors, which in turn improves the robustness of trained robot policies that rely on them.

One critical weakness of this work is that the diversity of human strategies is not analyzed and discussed within the main text. It's not clear for continuous problems (such as handover or sequential task) how to identify human strategies and then analyze them. The appendix includes further info but I believe as a core aspect of this work, the main manuscript should provide a better explanation on those details.

Second weakness is the experimental setup used for HRI scenarios. The user interface is based on a teleoperation interface (using mobile phones), which I assume would not provide representative human movements, and thus spoils their strategy selection and execution preferences.
A more representative teleoperation interface would make the experiments and results more convincing.

**Summary Of Recommendation:**

The paper proposes a novel extension of prior work which improves human-robot collaboration performance for three different tasks. This is achieved by incentivizing the encoding of diverse human strategies while learning robot policies. The experimental setup in terms of how the data was acquired remains to be a critical issue in order to justify their representativeness of real human movement and strategy selection behaviors.

---

> ### Author Response · Authors · 2021-08-23
> **Response to Reviewer RH4p**
>
> Thanks for your thoughtful review! We are pleased to know that you agree with our key insight that integrating diverse human strategies into the robot policy learning process would effectively improve the model robustness. You brought up some great questions and suggestions. Please find our answers below:
>
> - **Move details of diverse human strategies from appendix to the main text:** We agree with the suggestion and have placed more discussions in the main manuscript. We have added Line 206-208, Line 212-214, Line 216-219 in the revision to discuss the diverse human strategies in different task domains. We left additional details and an illustrative figure in the Appendix due to the page limit.
>
> - **The mobile teleoperation interface might spoil users’ strategy execution preferences:** We agree the teleoperation interface might affect user behavior patterns, but we still find a large diversity in strategies and execution preferences from different users with the same interface. So even though the interface might not be perfect, the challenge of collaborating with diverse human behaviors still presents, and this work aims to overcome this issue from the algorithm perspective. In the future, we are planning to use VR devices or motion capture systems to allow more natural ways to provide demonstrations.
>
> - **Difference between Co-GAIL and DIAYN in Fig.4:** In Fig.4(a), Co-GAIL (green dots) is the only algorithm that covers the left region (x<-0.6) of the given data distribution (red triangles), which proves its strong capability of discovering diverse human strategies. In contrast, DIAYN (purple dots) tends to perform handovers barely on the right side (x>-0.6) and explore those top areas (y>-0.075) that are out of the main distribution of reasonable human behaviors (red triangles). This is because the learning objective of DIAYN encourages the exploration “as diverse as possible” [32], which will result in unconstrained behaviors. This issue becomes more severe in high dimensional task domains like HR-Handover and HR-SeqManip. For example, in Fig. 4(b) we observe that Co-GAIL has a better separation in both X and Y dimension in two task domains, while DIAYN failed to separate different strategies in one of the dimensions (X-axis in HR-Handover and Y-axis in HR-SeqManip).
>
> - **Why do baseline methods perform better in Table 2:** The interpolation evaluation (Table. 2) is designed for analyzing the overfitting issue of the baseline methods (Line 276-278). The term "overfit" in this paragraph means the entire code space is mapping to a subset of strategy space, indicating that the algorithm fails to capture the diverse strategies from the given demonstrations. Because we uniformly sample the strategy code during training, the learned behaviors are distributed to the entire code space. The overfitting happens when few strategies are mapped to a large chunk of the code space redundantly, and other strategies are ignored. This is why the baselines such as GAIL tend to focus on generating a limited set of familiar strategies and ignore the more challenging cases with less training data coverage, contributing to their higher average success rate in Table 2. For example, in the visualization of the interpolation results on the 2D Fetch Quest (Fig. 3), the baseline methods fail to uncover all four types of strategies and prefer to generate strategies 3 and 4 for dataset $D_1$ (strategies 1 and 2 for dataset $D_2$), which are more common in the dataset. Similar phenomena can be observed in the Handover (Fig. 4a), where the baseline methods tend to conduct the handover on the right region that has more samples in the dataset. In contrast, Co-GAIL generates more diverse interactive behaviors, but at the cost of subjecting the policy to more challenging / less familiar strategies. We believe that this is a challenging research problem that might suggest future research on learning a wide set of behaviors with a single policy model. We plan to draw insights from multi-task policy learning into the future works.

---

> > ### Author Response · Authors · 2021-08-23
> > **Response (Part 2)**
> >
> > Please note that our response has been split into two parts due to space constraints. (Part 2/2)
> >
> > - **Details of the real-human evaluation:** The training data was collected with users that have experience with the teleoperation interface, including authors of the paper. In the pilot study with users, we invite four participants with no prior experience with the system or knowledge about robotics / HRI research. Before starting an experiment, each participant received a 5 minutes introduction of the system, a 10-minutes instruction on how to use the system, and another 15 minutes for the participants to familiarize themselves with the system. The set of possible collaboration strategies were given as suggestions to the users prior to the evaluation process. When the evaluation starts, the users are free to decide their strategy and motions to perform each task without any instructions or guidance. Participants didn't communicate with each other or with the experimenter during task execution. Each participant carried out 20 rounds of tasks for each task domain. In each round, the robot agent is controlled by one of the algorithms selected at random, and unknown to the user. More details of the training data collection can be found in Appendix C.
> >
> > - **Minor typos:** Thanks! We have fixed them in the revision.

---

> > > ### Comment · Reviewer_RH4p · 2021-09-01
> > > **Thanks**
> > >
> > > Thank you for the clarifications and updates on the paper. Although some of my concerns are alleviated, I still think the experimental setup is not ideal (surely acknowledging the covid restrictions). I keep my recommendation as 'weak accept'.

---

### Official Review · Reviewer_4Wjg · 2021-07-24

**Originality:** Good
**Technical Quality:** Very Good
**Clarity Of Presentation:** Good
**Impact:** 3

**Recommendation:**

Weak Accept: I recommend accepting the paper, but will not argue for my recommendation if the majority of other reviewers have a different opinion.

**Summary:**

This paper presents a technique to generate robot behavior that can collaborate successfully with humans by learning from human-human interaction. They model this as a two-agent imitation learning problem where behaviors for both the human and the robot are learned concurrently. Their main contribution is to extend the MA-GAIL [26] framework by introducing a latent representation of the human's strategy, also learned from the data, which enables the robot to adapt to the diverse set of behaviors exhibited by the human during training. They test this model in three collaborative tasks and show that it outperforms several baselines.

**Issues:**

- More details relating to the experiment design as well as the participants used for the study as described in the weaknesses.

- Space permitting, it would be good if authors can mention more detail about their experimental setup for reproducibility

- Is there a plan to release the software and data?

- Analyze and explain the results from table 2.

- The description of Fig 4b is difficult to understand.

- In 4.3, which learned strategy space is being used for Interpolation?

- line 24, not 30% in the HR-S

Typos/ minor issues:
- (line 116) an -> a
- (line 231) reply -> replay, if -> of
- (line 244) table 2-> 1
- (line 249) table 2-> 1


**Reviewer Expertise:**

Good: General knowledge of the area

**Strengths And Weaknesses:**

Strengths
 - The paper is well-written.
 - The technique is motivated by human behavior and grounded in past work.
 - Tested in multiple domains.
 - Experiments included their model interacting with humans which is rare.

Weaknesses:
- The experiment descriptions are missing some details like the user recruitment procedure including for the training data. Also, were the different strategies that the humans used pre-decided? Was any form of communication permitted between participants?
- The superior performance of the baselines in table 2 is explained in the paper with overfitting. Can you provide some more detail why that should be the case since Interpolation is not necessarily playing trajectories from the training data? If I understand correctly,  the interpolated trajectories should also contain multi-modal behavior that MA-GAIL should find difficult to adapt to. Also, would be helpful if it is explained how the strategy sampled here was learned.

That makes sense, seems to indicate that the proposed method underperformed the baselines. It would be useful if the authors presented some reasons for this.

**Summary Of Recommendation:**

The paper is well-written, they describe the problem clearly. Their solution seems logical given the explanation of past work. Their experiments provide a thorough test of their method.

---

> ### Author Response · Authors · 2021-08-23
> **Response to Reviewer 4Wjg**
>
> Thanks for your thorough and helpful review! We are happy to hear that you are pleased with our motivation inspired by real human behaviors and thorough experiment results to showcase the advantage of the proposed method. You also brought up some great questions and suggestions. Please find our answers below:
>
> - **Details of the pilot study:** The training data was collected with users that have experience with the teleoperation interface, including the authors of the paper. In the pilot study with users, we invite four participants with no prior experience with the system or knowledge about robotics / HRI research. Before starting an experiment, each participant received a 5 minutes introduction of the system, a 10-minutes instruction on how to use the system, and another 15 minutes for the participants to familiarize themselves with the system. The set of possible collaboration strategies were given as suggestions to the users prior to the evaluation process. When the evaluation starts, the users are free to decide their strategy and motions to perform each task without any instructions or guidance. Participants didn't communicate with each other or with the experimenter during task execution. Each participant carried out 20 rounds of tasks for each task domain. In each round, the robot agent is controlled by one of the algorithms selected at random, and unknown to the user. More details of the training data collection can be found in Appendix C.
>
> - **Why do baselines perform better in the interpolation evaluation and why is it explained as overfitting:** The "overfit" in this paragraph means the entire code space is mapping to a subset of strategy space, indicating that the algorithm fails to capture the diverse strategies from the given demonstrations. Because we uniformly sample the strategy code during training, the learned behaviors are distributed to the entire code space. The overfitting happens when few strategies are mapped to a large chunk of the code space redundantly, and other strategies are ignored. This is why the baselines such as GAIL tend to focus on generating a limited set of familiar strategies and ignore the more challenging cases with less training data coverage, contributing to their higher average success rate in Table 2. For example, in the visualization of the interpolation results on the 2D Fetch Quest (Fig. 3), the baseline methods fail to uncover all four types of strategies and prefer to generate strategies 3 and 4 for dataset $D_1$ (strategies 1 and 2 for dataset $D_2$), which are more common in the dataset. Similar phenomena can be observed in the Handover (Fig. 4a), where the baseline methods tend to conduct the handover on the right region that has more samples in the dataset. In contrast, Co-GAIL generates more diverse interactive behaviors, but at the cost of subjecting the policy to more challenging / less familiar strategies.  We believe that this is a challenging research problem that might suggest future research on learning a wide set of behaviors with a single policy model. We plan to draw insights from multi-task policy learning into the future works.
>
> - **Plan for code release:** Yes, we will publicly release the code as well as the demonstration dataset.
>
> - **The description of Fig 4b is hard to understand:** Thanks for mentioning this point. We have revised this part in the manuscript to remove the ambiguity. In this experiment, we replay the entire unseen testing demonstration dataset and visualize the latent code estimation of each trajectory outputted by the strategy recognition network $\psi$. Since the dimension of the latent code we use in this work is 2, the visualization is a 2d scatterplot. The color of each dot is the ground truth strategy category of the trajectory (used only for visualization purposes). Codes for the same strategy tend to cluster in the same region. We can see better separation of latent strategy learned using our approach than the other two methods.
>
> - **In 4.3, which learned strategy space is used for interpolation:** The strategy space here is referred to the input strategy code $z$. We followed prior works infoGAIL[25] and infoGAN[45] to uniformly sample the code space during the training and interpolation process. During the interpolation, we uniformly sweep over the code space to generate different collaborative strategies.
>
> - **Typos / minor issues:** Thanks! We have fixed them in the revision.

---

> > ### Comment · Reviewer_4Wjg · 2021-09-04
> > **Thanks**
> >
> > I appreciate the detailed clarifications. My recommendation remains the same (weak accept).

---

### Official Review · Reviewer_qTpv · 2021-07-27

**Originality:** Very Good
**Technical Quality:** Excellent
**Clarity Of Presentation:** Excellent
**Impact:** 4

**Recommendation:**

Strong Accept: I recommend accepting the paper and will argue for my recommendation even if other reviewers hold a different opinion.

**Summary:**

This paper presents a method for learning two models for human-robot collaborative tasks: one that reproduces human-like behaviors, and one that optimizes the robot's behavior for efficient collaboration. In learning the former, the human's strategy is represented as a latent representation and used to produce trajectories that maximize informativeness. The results indicate that the proposed method produces trajectories that reflect a variety of strategies demonstrated by a human (even when those strategies are unbalanced in the dataset). Furthermore, the proposed method resulted in more successful handovers during a real robot task.


**Issues:**

* More discussion on the effect of the hyper parameters in eq 4 would be useful.
* During training, what defines the ground truth for z in eq 2?

**Reviewer Expertise:**

Good: General knowledge of the area

**Strengths And Weaknesses:**

Strengths:
* The paper is well written and easy to follow.
* The motivation is convincing, and the related works section clearly explains why the proposed method provides a benefit over current SOTA.
* The evaluation is very thorough as a result of comparing the proposed method to four different baseline approaches.
* The results themselves are compelling, and test each part of the system (latent space learning, policy learning, and execution).

Weaknesses:
* More discussion on the effect of the hyperparameters in eq 4 would be useful.
* During training, what defines the ground truth for z in eq 2?
* Axis labels in Figures 3 and 4 are way too small

**Summary Of Recommendation:**

Overall, this paper is clear and presents thorough, compelling results.

---

> ### Author Response · Authors · 2021-08-23
> **Response to Reviewer qTpv**
>
> Thanks for your thoughtful comments! We are pleased to hear that you found our motivation convincing and the experiment results thorough and compelling. You brought up some great questions and suggestions. Please find our answers below:
>
> - **Discussion on the effect of hyperparameters in eq 4:** Thanks for the suggestion! We have added a section in the appendix (Line 491-496) related to the effect of hyperparameters. In all three experiments, we used $\lambda_1 = \lambda_2 = 0.1$. We found that large $\lambda_1$ and $\lambda_2$ would cause unstable training procedures of the GAIL algorithm, where the objectives that measure the reconstruction error of the strategy code and human actions in eq. 4 dominate the loss function and cause the model to neglect the first GAIL learning objective. We empirically found $\lambda_1 = \lambda_2 = 0.1$ achieve the most stable learning performance in all three task environments.
>
> - **During training, what defines the ground truth for $z$ in eq 2:** During the training process, the strategy code is uniformly sampled (Main paper line 154) and used as the ground truth in eq 2. This is a widely used code sampling method in prior works such as infoGAIL[25] and infoGAN[45] for learning diverse policy/image generation. The details of the sampling process can be found in Appendix. B.1.
>
> - **Size of the axis labels in the Figures:** Thanks and we have fixed them in the revision.

---

> > ### Comment · Reviewer_qTpv · 2021-09-03
> > **Response**
> >
> > Thank you for the clarifications. My recommendation remains the same (accept).

---

### Meta-Review · Area_Chair_2Z6f · 2021-08-12

**Recommendation:** Accept (Poster)
**Confidence:** 4

**Metareview:**

The paper received 4 reviews which are on average leaning slightly towards an accept. The paper is well written and well motivated, the evaluation is thorough. The paper heavily builds on prior work, result in rather low impact/contribution scores. Some details need to be clarified. The most important concerns circle around the experiments. Some of the results (in particular why the proposed method did worse in one setting) need a deeper analysis. Related to that there are also some doubts about the validity of the setup and results for some of the experiments.

## Update after author response and AC & reviewer discussion
The reviewers appreciated the detailed responses and additional experiments and analysis. They are now all in favor of accepting the paper.

---

> ### Author Response · Authors · 2021-08-23
> **Response to Area Chair 2Z6f**
>
> We thank all the reviewers and the meta reviewer for their valuable comments! We have revised the paper as suggested by the reviewers. We address specific questions in the meta-review below:
>
> - **Potential impact/contributions:** To the best of our knowledge, this work is the first to point out that a critical challenge of learning an effective human-robot collaboration policy is to model the diversity in human collaboration behaviors, including variations in both high-level strategies and low-level movements. Co-GAIL addresses this challenge by introducing a co-optimization process that combines modeling diverse human strategies and training assistive robot policy into a unified learning framework. While prior works such as InfoGAIL[25] and MA-GAIL[26] use the GAIL framework for multi-agent problems, they exclusively focus on proficiency in completing a task and do not explicitly address the challenge of modeling diverse collaboration behaviors which becomes key for interaction with humans. Technically, our work introduces a novel learning method (objective and training regime) for diversity exploration, and an on-line human strategy recognition module. Together, these contributions showed for the first time proficiency at interaction with real users, including three different domains. Co-GAIL can capture and learn to react to real human collaboration strategies, and it outperforms prior works in both procedural evaluation and online human-in-the-loop evaluation.
>
> - **Performance of the proposed method in interpolation evaluation:** The interpolation experiment offers insights into an algorithm’s ability to learn diverse strategies from the given dataset. We observed in the visualized interpolation results (Figure 3 and 4(a)) that the baseline methods tend to focus on generating a limited set of familiar strategies and ignore the more challenging cases with less training data coverage. The success rate metric reported in Table 2 indirectly supports the observation. The baseline methods receive higher success rates in this metric than Co-GAIL. This is because mapping the entire code space to a subset of the strategy distribution allows the policy model to focus its learning capacity on the few strategies with more training data. In contrast, Co-GAIL generates more diverse interactive behaviors, but at the cost of subjecting the policy to more challenging / less familiar strategies. Finally, Co-GAIL achieved the best performance in both unseen demonstration replay evaluation and real-human evaluation, further supporting that Co-GAIL generates more diverse and realistic strategies for human-robot collaboration.
>
> - **Evaluation protocol:** We have reported results with additional random seeds, added more participants to the human-in-the-loop evaluation, and provided further details for the pilot study.

---

### Decision · Program_Chairs · 2021-09-13

**Decision:**

Accept (Poster)

**Comment:**

The paper received 4 reviews which are on average leaning slightly towards an accept. The paper is well written and well motivated, the evaluation is thorough. The paper heavily builds on prior work, result in rather low impact/contribution scores. Some details need to be clarified. The most important concerns circle around the experiments. Some of the results (in particular why the proposed method did worse in one setting) need a deeper analysis. Related to that there are also some doubts about the validity of the setup and results for some of the experiments.

## Update after author response and AC & reviewer discussion
The reviewers appreciated the detailed responses and additional experiments and analysis. They are now all in favor of accepting the paper.